# Mucins as Potential Biomarkers for Early Detection of Cancer

**DOI:** 10.3390/cancers15061640

**Published:** 2023-03-07

**Authors:** Shailendra K. Gautam, Parvez Khan, Gopalakrishnan Natarajan, Pranita Atri, Abhijit Aithal, Apar K. Ganti, Surinder K. Batra, Mohd W. Nasser, Maneesh Jain

**Affiliations:** 1Department of Biochemistry and Molecular Biology, University of Nebraska Medical Center, Omaha, NE 68198, USA; 2Fred & Pamela Buffett Cancer Center, Eppley Institute for Research in Cancer and Allied Diseases, University of Nebraska Medical Center, Omaha, NE 68198, USA; 3Division of Oncology-Hematology, Department of Internal Medicine, VA Nebraska Western Iowa Health Care System, University of Nebraska Medical Center, Omaha, NE 68105, USA

**Keywords:** mucin biomarker, early cancer detection, liquid biopsies, exosomes, autoantibodies

## Abstract

**Simple Summary:**

Early cancer detection is a challenge in treating cancer patients. Remarkably, carcinomas of the breast, lung, liver, pancreas, ovary, and colon contribute to more than 50% of total incidences and mortality annually, which could be reduced by early detection. As mucins are highly upregulated during the early progression of these cancers, several studies have explored mucins as potential biomarkers. Due to their membrane-bound and secretory nature, mucins have been detected in tumor biopsies and liquid biopsies of cancer patients, including blood and urine samples. We compiled here previous studies advocating for the use of mucins as potential biomarkers for the early detection of cancer and discuss the opportunities and challenges related to the use of mucin biomarkers in combination with other biomarkers and detection modalities.

**Abstract:**

Early detection significantly correlates with improved survival in cancer patients. So far, a limited number of biomarkers have been validated to diagnose cancers at an early stage. Considering the leading cancer types that contribute to more than 50% of deaths in the USA, we discuss the ongoing endeavors toward early detection of lung, breast, ovarian, colon, prostate, liver, and pancreatic cancers to highlight the significance of mucin glycoproteins in cancer diagnosis. As mucin deregulation is one of the earliest events in most epithelial malignancies following oncogenic transformation, these high-molecular-weight glycoproteins are considered potential candidates for biomarker development. The diagnostic potential of mucins is mainly attributed to their deregulated expression, altered glycosylation, splicing, and ability to induce autoantibodies. Secretory and shed mucins are commonly detected in patients’ sera, body fluids, and tumor biopsies. For instance, CA125, also called MUC16, is one of the biomarkers implemented for the diagnosis of ovarian cancer and is currently being investigated for other malignancies. Similarly, MUC5AC, a secretory mucin, is a potential biomarker for pancreatic cancer. Moreover, anti-mucin autoantibodies and mucin-packaged exosomes have opened new avenues of biomarker development for early cancer diagnosis. In this review, we discuss the diagnostic potential of mucins in epithelial cancers and provide evidence and a rationale for developing a mucin-based biomarker panel for early cancer detection.

## 1. Introduction

Early detection correlates with better survival in cancer patients [1]. However, clinically validated biomarkers are not available for the early detection of highly lethal malignancies, such as pancreatic, breast, lung, prostate, liver, and colorectal cancers, which contribute to more than 50% of cancer-related mortalities in the USA [2,3,4]. On the other hand, the survival rates in some cancers, such as breast, ovarian, and prostate cancers, have appreciably improved, mainly due to the advances in screening methods, biomarker development, and surgical procedures [2,3]. As biomarker-based screening is non-invasive and cost-effective, it is a preferred method for early cancer detection. However, achieving clinically acceptable sensitivity and specificity of biomarkers remains an ongoing challenge in most cancers, necessitating the development of new biomarkers and their use in combination with currently available screening methods [5,6]. Blood and other body fluids are the most common sources of specimens for biomarkers such as microRNAs, circulating tumor cells (CTCs) and circulating tumor DNAs (ctDNA), autoantibodies, cytokines, exosomes, and other secretory molecules, including mucins [7,8,9,10,11,12].

Mucins are high-molecular-weight glycoproteins that are progressively deregulated from early to advanced stages of different epithelial cancers. Therefore, mucins have been widely explored as biomarkers for early cancer detection and disease prognosis [13,14,15,16,17]. Moreover, the expression analyses of unique mucin signatures and their combination with clinically practiced imaging techniques could be used to better diagnose malignant tissue growth at the early stages. Therefore, in this review article we attempt to determine the potential of mucin-based biomarkers in early-stage cancer diagnosis, which could guide the future development of potential biomarker panels. Moreover, early cancer detection allows better treatment options for cancer patients, which includes surgical intervention. Previous studies have reported that early cancer detection followed by surgical intervention and adjuvant chemotherapy could improve patient survival remarkably in most solid malignancies [18,19,20,21,22,23].

We review the current literature on the analysis of mucins as biomarkers for the early detection of major epithelial cancers that contribute significantly to cancer-related deaths and are also known for predominant mucin deregulation, such as lung, colon, pancreas, prostate, liver, breast, and ovary cancers. We discuss the clinicopathological significance of deregulated mucins in the aforementioned malignancies, their association with non-mucin biomarkers, and describe their utility in early detection and prognosis. We also highlighted recent studies demonstrating the detection of mucins in extracellular vesicles (EVs) and discuss the role of emerging technologies for ultrasensitive detection of mucins in cancer. 

## 2. Mucin Deregulation in Cancer

Besides oncogenic mutations, tissue-specific factors regulate tumor progression and aggressiveness. These tissue-specific factors include disease-specific proteins, metabolites, and immunologically active products such as antibodies, immune cells, and cytokines [24,25,26]. Disease-specific proteins, metabolites, and antibodies can be quantified to assess the oncogenic progression and, therefore, can be utilized to develop biomarkers for cancer detection. High-molecular-weight mucin glycoproteins are a major component of most epithelial linings, such as the respiratory, gastrointestinal, and reproductive tracts, providing a protective barrier from infections and harsh physiological cues [27,28]. The two major classifications of mucins are (A) gel-forming and non-gel-forming mucins and (B) membrane-bound and secretory mucins [29]. These mucins differ in size, domain structures and organizations, glycosylation, and functions [28,30,31,32,33]. Epithelial malignancies are often characterized by deregulated mucin expression [31,32] and their aberrant glycosylation pattern [34,35]. Concomitant to oncogenic mutations, a loss or gain of mucin expression has been reported in different cancers, including carcinomas of the pancreas, colon, lung, ovary, and breast [31,36,37,38]. For example, MUC2, MUC4, MUC5AC, and MUC16 are absent in the normal pancreas. However, the oncogenic transformation and the pathological cues trigger expression from these mucins as early as the PanIN stage of pancreatic cancer [36,39]. Similarly, in the case of colon cancer, loss of MUC2 and MUC4 and gain of MUC5AC and MUC16 expression have been reported temporally with oncogenic progression [40,41,42]. Significant deregulation in mucin expression has also been observed in other epithelial malignancies, including ovarian, breast, and gastric carcinomas [31,38,43,44]. In addition to oncogenic mutations, other tumor-associated factors, such as hypoxia, immunosuppressive cytokines and chemokines, and tumor-specific metabolites, have been reported to contribute to the altered mucin profile [45,46,47,48]. Furthermore, epigenetic factors such as DNA methylation are considered to play a critical role in altering the mucin expression profile. A previous study reported that a change in the CpG methylation status of MUC1 significantly alters its expression, leading to a pathological role of MUC1 in different cancers [49]. We previously reviewed the promoter organization and genetic elements contributing to mucin expression and deregulation. For instance, secretory mucins contain a TATA box upstream to the transcription initiation site, and transcription factors such as Sp1 and Sp3 have been reported to be involved in the induction of mucin expression in various cancers [30]. Pathologically, an altered mucin profile in epithelial malignancies has been correlated with poor prognosis and survival of cancer patients. For instance, high MUC4 expression correlates with poor prognosis and survival in lung and pancreatic cancer patients [50,51]. Similarly, elevated MUC16 expression is a predictor of disease progression as well as poor survival in ovarian cancer patients [52]. However, a moderate or diffuse expression of MUC16 has been found to be strongly associated with poor survival in PDAC patients (n = 200 patients), esophageal adenocarcinoma (n = 95 patients), and gastric adenocarcinoma (n = 119 patients) [53]. In contrast, a focal expression of MUC16 has been reported to correlate with better clinical outcomes in colorectal adenocarcinoma (n = 39) [53]. Similarly, the deregulation of secretory mucin MUC5AC expression correlates differently with survival and disease prognosis in various cancers. For example, in gastric carcinoma, a decrease in MUC5AC expression correlates with poor prognosis, whereas in pancreatic, colon, and lung cancers, an increase in MUC5AC correlates with poor prognosis [54,55,56,57,58]. Deregulated expression of other mucins, such as MUC1, MUC2, MUC3, MUC6, MUC17, and MUC20, has also been reported in different cancers, either in their native or post-translationally modified forms [59,60,61,62] (Figure 1). Thus, deregulation in mucin transcripts and their post-translational modifications alter their functions distinctly in each cancer type. Based on their altered expression, mucins have been assessed for their potential as biomarkers for early diagnosis and disease prognosis (Figure 1 and Figure 2).

## 3. Mucins as Cancer Biomarkers

Early detection of cancer is the key to improving the clinical management and survival of cancer patients [63,64,65,66]. The cancers associated with poor survival rates often lack methods for early detection. Therefore, an ongoing challenge in the clinical management of these cancers is to identify and validate novel biomarkers for their early detection. In this regard, mucins are considered potential biomarker candidates for early cancer detection [31]. Herein, we discuss the utility of mucins as biomarkers in major malignancies that are associated with poor survival. The top malignancies in females in terms of incidences and mortality include lung cancer, breast cancer, pancreatic cancer, ovarian cancer, and colorectal cancer, which account for 56% of estimated annual deaths in the USA. Similarly, lung, colorectal, pancreatic, prostate, and liver cancer contribute to more than 50% of cancer-associated deaths in men annually [3]. All these solid malignancies have been studied thoroughly for altered mucin expression to understand their functional role in cancer pathogenesis and examine their potential as targets for diagnostics and therapy. In the following section, we summarize studies focused on examining the utility of cancer-specific mucins as biomarkers in each of the abovementioned malignancies. 

## 4. Lung Cancer Diagnosis and Mucin Biomarkers

Lung cancer (LC) remains a major contributor to cancer-related mortalities, with an estimated 236,740 new cases and nearly 130,180 deaths due to lung and bronchus cancer in 2022 in the United States [3,67]. LC is mainly subdivided into non-small-cell lung cancer (NSCLC), which accounts for nearly 80–85% of total LC cases, and small-cell lung cancer (SCLC), which accounts for nearly 13–15% of LC cases diagnosed [3,68]. For NSCLC, surgery is the best treatment option for the early-stage patients (stage I–II), and the five-year survival rate for early-stage patients is 60–63%, whereas it is only 25–28% in SCLC. Significant developments have been seen toward the development of treatment modalities, such as targeted therapies and immunotherapy [69,70,71]. Unfortunately, late diagnosis is one of the factors associated with a high mortality rate in LC patients, resulting in a five-year survival rate of 7% for advanced NSCLC and 3% for extensive-stage SCLC [72,73]. The major risk factor for LC is smoking, accounting for approximately 80% of all LC cases, with smokers having a 20-fold higher risk of developing LC than non-smokers [74,75]. Nearly 65–70% of SCLC cases and more than 50% of NSCLC cases are diagnosed at a stage when tumors have metastasized beyond the lung, with systemic metastases in >50% of patients diagnosed with LC [73,75]. One of the major problems associated with the high lethality of LC is the lack of early detection biomarkers and screening methods [72]. 

The screening methods available for LC detection/diagnosis as per the recommendations of the US Preventive Services Task Force (USPSTF) guidelines include low-dose computed tomography (LDCT); the guidelines recommend yearly screening for LC using LDCT scanning for people (i) aged between 50 and 80 years in fairly good health, (ii) people currently smoking or people who have quit within the past 15 years, and (iii) people who have a smoking history of at least 20 packs/year [72,73]. The screening should be stopped once a person has not smoked for more than 15 years or develops other health problems substantially limiting their life expectancy or capability to have healing lung surgery. The anticipated benefit of this screening was the reduction in LC-associated deaths, though it was cautioned that not every person who gets screened will benefit [76]. It was reported that LDCT screening is not enough to detect all LC cases; secondly, not all the detected cancers are early-stage disease [72,77,78]. Moreover, not all CT scan abnormalities represent cancer, and therefore, such patients need other invasive tests, such as biopsies [78,79]. To improve the diagnostic ability of low-dose CT scans, efforts have been directed toward developing potential non-invasive or less invasive methods, such as the detection of cancer-specific molecules or antigens in the circulatory system, body fluids, or small tissue biopsies [80]. 

Multiple studies have demonstrated the potential of various mucins for the early detection of LC [37,81,82,83]. In a normal human lung, the goblets cells of the respiratory apical epithelium produce and release mucins for the protection of the respiratory airways and alveoli from dehydration, pathogen entry, injuries, and other physiological or non-physiological chemical agents [84,85,86]. The expression of MUC1 is very weak in the tracheobronchial epithelium and undetectable in the submucosal glands and bronchioles [87]. Similarly, MUC2 is also weakly expressed in the majority of lung tissues, except for a moderate expression at the basal pole of some goblet cells [87]. The expression of MUC4 is observed in the ciliated bronchial cells following 12 weeks of gestation [87,88,89]. At the same time, the expression of MUC5AC is observed during the 13th week of pregnancy. At this stage, MUC5AC is expressed moderately in segmental bronchi, and afterward, a strong expression of MUC5AC is observed in the trachea and bronchi. In addition, MUC5AC is present in the surface epithelium of goblet cells and glandular ducts, but there is no MUC5AC expression observed in small alveolar epithelial cells or bronchioles [90,91]. The expression of MUC5B is observed following the 13th week of gestation in the mucosa of the submucosal gland, glandular duct, and bronchial epithelium, whereas it is not expressed in the epithelial cells of alveoli and bronchioles [37,91]. Similarly, MUC7 expression is observed in some of the submucosal gland’s serous cells, and a high expression of MUC13 has been reported in the trachea [92]. The expression of MUC16 or CA125 was reported in the goblet and mucous cells of the submucosal gland as well as on the human tracheal tissue (epithelial surface). MUC16 was also observed in ‘normal’ respiratory tract mucus and normal human bronchial epithelial (NHBE) cell secretions [93]. The expression of other mucins, such as MUC3 and MUC6, has not been observed in normal respiratory mucosa [89]. In summary, mucin genes follow a unique and differential expression pattern during lung development and in the different regions of the respiratory airway mucosa.

The dynamicity of mucin expression and post-translational modifications such as glycosylation in lung tissues prompted researchers to investigate the utility of these mucins as biomarkers for the early detection of LC [88]. The variations in the expression level and pattern of mucins mRNA in LC (adenocarcinoma and carcinoma) cell lines and tissues have been reported using multiple techniques, such as Northern blot, dot blot, and immunochemical methods [88,94]. Interestingly, mucin expression was found to be correlated with the different stages of LC, including hyperplasia, dysplasia, and squamous metaplasia. During epithelial hyperplasia, the expression pattern of mucin genes was analogous to normal mucosal cells, whereas the quantitative analysis showed a substantial variation in MUC4 and MUC5AC expression. For instance, MUC5AC was found to be absent in basal cells, but it was highly overexpressed in the upper parts of the epithelium in basal cell dysplasia, squamous dysplasia, hyperplasia, metaplasia, and goblet cell hyperplasia. On the other hand, MUC4 was found to be overexpressed in squamous metaplasia and dysplasia [88,89]. Pan et al. showed a significant expression of MUC1 in the exosomes secreted by NSCLC cells and in the plasma of NSCLC patients [95]. The proteomic analysis of proteins purified from exosomes of the NCI-838 NSCLC cell line and exosomes from the plasma of LC patients showed that MUC1 is highly expressed (>8-fold) in exosomes isolated from the cell lines and plasma compared to total cellular membrane proteins and exosomes isolated from healthy individuals [95]. The outcomes of the study suggest that MUC1 is selectively enriched in the exosomes of NSCLC patients, putting forward the importance of this mucin for detecting LC. Furthermore, polarized MUC1 expression was found to decrease with tumor progression from adenomatous hyperplasia to mixed subtypes (adenomatous hyperplasia, bronchioalveolar carcinoma, and adenocarcinoma), whereas the expression of depolarized MUC1, MUC2, MUC5AC, and MUC6 increased with such progression [96,97,98]. In addition, a correlation was observed between p53 abnormalities and an elevated expression of depolarized MUC1, MUC5AC, and MUC6 [96]. Mucinous adenocarcinoma (a more aggressive LC subtype) showed higher expression of secretory mucins, including MUC2, MUC5AC, and MUC6 [99,100]. Recently, Kishikawa et al. performed molecular, immunohistochemical, and clinicopathological evaluations of 70 invasive mucinous adenocarcinoma tumor samples and reported a patchy or diffuse expression of various mucins, including MUC1, MUC2, MUC4, MUC5AC, and MUC6 [99]. Interestingly, patients with a diffuse expression of MUC6 had a favorable outcome compared to those expressing other mucins. Moreover, MUC6 expression was found to be associated with wild-type KRAS, hence characterizing a distinct LC subset [99]. This suggests that the evaluation of genomic alterations and secreted mucins in the patient sera and other biopsies could be a promising tool for detecting LC. 

A recent study showed the presence of secreted MUC16 (CA125) in the culture supernatant of NSCLC cell lines and patient samples [101]. Some studies utilized combinatorial approaches based on mucin expression and other LC-specific antigens. For example, the expression of MUC5AC is an indicator of normal bronchial cells (fully differentiated) [102], and thus can be used to detect the presence of bronchial cells in the circulation. In addition, cancer-specific markers such as annexin-II and mucins such as MUC1 can be used as combination markers to detect LC cells in the circulation. Two interesting recent studies demonstrated the utilization of MUC1 for subtype identification and non-invasive detection of SCLC [103,104]. High expression of MUC1 has been reported in SCLC patient blood-derived CTCs and tissue samples [104].

A comparative evaluation of one or more available biomarkers, such as CEA, CYFRA 21-1, TAG72-3, neuron-specific enolase (NSE), and squamous cell carcinoma antigen (SCC), along with the MUC1/CA15-3 and MUC16/CA125 mucins, has provided a platform for the early detection of NSCLC and its differentiation from SCLC [105]. Thus, a higher expression of CA125, CA15-3, CEA, CYFRA21-1, SCC, and TAG72-3 in the serum samples of patients is considered an early detection biomarker of NSCLC. Further, the high serum expression of CEA, TAG72-3, CA15-3, and CA125 represents adenocarcinoma, whereas high SCC, CEA, and CYFRA21-1 identify squamous carcinoma [105]. Another similar combinational serum-based biomarker study involving the analysis of CA125, CEA, CY211, SCC, and NSE established the role of high serum levels of CA125 and CEA in the screening of NSCLC [106]. In addition to LC screening, high levels of MUC1, CA125, KL-6, CYFRA21-1, and LAMC2 also predict the overall survival of NSCLC patients, thereby demonstrating the prognostic implications of mucins in NSCLC [107,108]. Studies evaluating mucin-based serological screening in LC have been summarized in Table 1-A. These studies suggest evaluation of mucin expression in combination biomarker panels may enhance the diagnostic performance. Overall, the detection of mucins (secretory as well as membrane-bound) in different types of biopsies (liquid to small tissue) in combination with available biomarkers provides a potential avenue to detect LC in the early stages, which can go undetected by conventional screening methods such as LDCT.

## 5. Breast Cancer Early Detection and Mucin Biomarkers

Despite the improved detection methods and available treatment options that have substantially improved the clinical outcomes [139,140], breast cancer (BC) is among the leading causes of cancer-related deaths in women worldwide [3,141]. BC is the most frequently diagnosed cancer, with nearly 2,261,419 new cases in 2020 and 684,996 deaths worldwide in 2020 [2]. In the United States, the estimated number of new cases of BC in 2022 was 287, 850, resulting in 43,250 deaths [142]. Multiple screening methods are available to detect BC, including mammograms, breast ultrasounds, and breast MRIs. In addition, newer and experimental breast imaging tests for BC are also available, including breast tomosynthesis (3D mammography), molecular breast imaging or scintimammography or breast-specific gamma imaging, contrast-enhanced mammography, electrical impedance imaging, and elastography [143,144]. Mammography is the most commonly used method for early BC screening [140]. The clinical guidelines recommend the screening of women over the age of 50 years, which accounted for >42 million yearly screening exams [140,144,145,146]. This high-scale screening has facilitated the detection and treatment of BC at the early stages when tumors are localized. 

Although regular mammogram screening helps in the detection of BC at an early stage, it is likely that the mammogram misses some of cancers or overdiagnoses them [147,148]. Secondly, the interpretation of mammography images remains challenging, and thus the accuracy is subject to interobserver variability [147,149]. Recently, digital breast tomosynthesis or three-dimensional mammography has become a more common method. Still, its availability is restricted to a few centers, and the cost is remarkably high [145,150,151]. Therefore, there is a need to identify and develop alternative methods for the early detection of BC that may help to tackle the challenges of currently available detection methods. 

Mucins are significantly altered during different stages and subtypes of BC [35,152,153,154]. However, it is important to classify mucinous and non-mucinous carcinomas before analyzing the BC mucin profile. Earlier, a study showed that in a BC patient cohort, MUC2 was expressed in all mucinous carcinomas, 11.2% of invasive ductal carcinomas, and none of the invasive lobular and medullary carcinomas [155], whereas MUC1 was expressed in invasive BC, but not in medullary carcinomas. This study suggested that mucin expression correlates with BC origin and subtypes and therefore, mucins and mucin-like glycoproteins can be utilized as biomarkers for BC diagnosis and patient stratification. A previous study showed that mucin-like carcinoma-associated antigens (MCAs), CA15.3 (secretory/soluble fragment of MUC1), and CEA could be used for the early detection of BC and its metastatic progression [156]. Interestingly, an elevated level of these markers preceded a clinical diagnosis of metastases in high-risk BC patients. Later, the role of MUC1 in BC was described by Kufe et al. [157], who reported that MUC1 undergoes autocleavage and releases the N-terminal domain of MUC1 (MUC1-N) in the circulation or body fluids [158]. The loss of apical-basal polarity in breast epithelial cells leads to the overexpression of MUC1, and from the BC cells, MUC1-N is shed into the circulation, which provides the basis of MUC1-based screening in the body fluids of BC patients [153,159]. MUC1-N was found at a higher level in the plasma of BC patients. Further, BC patients with metastatic disease have a higher level of circulating MUC1 than patients with primary BC [159,160,161]. It was found that elevated circulating MUC1 is also useful in distinguishing BC from other cancers, such as hepatoma and ovarian carcinoma [159]. 

Interestingly, transformed MUC1 (tMUC1, a hypo-glycosylated form of MUC1) is a specific antigen that is overexpressed in the early stages of triple-negative BC (TNBC) patients [111,162]. Nearly 95% of TNBC tissues show positive staining for tMUC1, and the use of tMUC1-specific antibodies (TAB004) is also useful for the detection of circulatory MUC1 irrespective of breast tissue density. In this regard, a pilot study performed with banked serial samples demonstrated that the tMUC1 level was significantly increased even before two years of diagnosis [111]. This suggests that tMUC1 could be used as an early detection tool for TNBC patients, even in women with higher breast density where early-stage tumors are missed in mammography. Recently, MUC1-based theranostic approaches have also been developed using humanized TAB004 antibodies for detecting and treating early-stage TNBC [163]. Aptamer-based biosensing approaches with significant sensitivity have also been developed to detect MUC1 in the serum or circulation of BC patients [164,165]. 

Rakha et al. comprehensively analyzed the expression of MUC1, MUC2, MUC3, MUC4, MUC5AC, and MUC6 in 1447 cases of invasive BC and demonstrated the utility of MUC1 and MUC3 in predicting early metastasis [166]. Overexpression of MUC1 and MUC3 was reported in 90% of BC cases and found to be associated with high local recurrence and lymph node metastasis [166], suggesting the application of a MUC1-MUC3 detection panel for the prediction of local recurrence and lymph node metastasis. Expression of MUC1, MUC2, MUC5AC, and MUC6 was also observed in the early stages of BC patients [167,168]. The secreted mucins, MUC2 and MUC5B, were found to be associated with the aggressiveness of BC cells [169,170,171]. Similarly, overexpression of MUC1 and MUC6 was also reported in ductal hyperplasia [167], and overexpression of MUC2 was reported in the early stages of lobular carcinoma [168]. In a recent analysis, multiple mucins, including MUC1, MUC2, MUC5AC, and MUC6, were found to differentiate primary BC from metastatic BC, where MUC1 correlated with the low or early grade, MUC5AC with metastatic disease, and MUC6 emerged as an indicator of poor prognosis [172,173]. Analysis of MUC1 in combination with CEA and chemerin in the serum samples of BC patients resulted in higher sensitivity and accuracy for the early diagnosis compared to CEA or CEA in combination with chemerin [109]. Similarly, evaluation of MUC1 (CA15-3) in combination with CEA and soluble intercellular adhesion molecule-1 (sICAM-1) in BC patient sera was found to be useful for the early detection of lymph node and distant organ metastasis [110]. The analysis of individual mucins or a panel of mucins alone or in combination with other markers in the circulation, body fluids, and small tissue biopsies is a useful tool for early detection or screening even in the cases where routine mammography fails to yield satisfactory results (Table 1-B, Figure 2).

## 6. Role of Mucins in the Early Detection of Ovarian Cancer

Ovarian cancer (OC) is the fifth leading cause of cancer-related mortality among women in the United States [2]. Survival in OC patients largely depends on the histological type and stage of diagnosis of OC. Up to 90% of OCs are of epithelial origin; these are sub-classified into histological subtypes, such as serous, endometrioid, mucinous, clear cell carcinoma, and transitional tumors [174,175]. Among the various subtypes, high-grade serous carcinoma is the most aggressive. Early-stage detection and aggressive treatment are considered critical to the prolonged survival of OC patients [176,177]. Currently, CA125 (MUC16) is the most widely used serum-based mucin biomarker for OC [178,179]. Several studies have demonstrated that the serum CA125 level (>35 U/mL) increases during the early stages of OC [179]. All patients with CA125 levels above the cut-off are considered vulnerable for the development of OC and recommended for regular follow-ups. However, the limited sensitivity of CA125 detection remains a challenge for early-stage OC screening [180,181]. Therefore, CA125 has been evaluated with other biomarkers and diagnostic modalities and incorporated into diagnostic algorithms. These include human epididymis-4 (HE-4) antigen, chemokines, microRNAs, Risk of Ovarian Malignancy Algorithm (ROMA) index analysis, and clinically practiced imaging approaches such as ultrasound imaging and computed tomography (CT) [182,183,184,185,186,187,188,189]. Previously, CA125, combined with M-CSF and OVX1, enhanced sensitivity of the panel as compared to a single biomarker for the detection of all histotypes, and the biomarker panel was effective in diagnosing all stages, including early-stage OC [190]. Later, in another study, serum specimens were analyzed for four tumor markers, including MUC16, CA72-4, CA15-3, and M-CSF, in healthy women (n = 100), benign ovarian carcinoma (n = 45), and invasive epithelial ovarian carcinoma (N = 55) to evaluate biomarker potential in combination with artificial neural network (ANN) analysis [191]. Interestingly, multiple marker analysis was more effective in diagnosing early-stage OC compared to a single marker, as analyzed with the help of the ANN-derived index. Similarly, other serum components, such as OPN, IL8, MIF, HE4, and CA72-4, have been investigated in combination with CA125, with encouraging readouts, suggesting that the CA125-based combination biomarker panels enhance the sensitivity of early detection of OC [113,192]. Recently, Dochez et al. showed that the combination of HE4 and CA125 is an efficient biomarker tool for OC diagnosis [186]. The area under the curve for the combination of CA125 and HE4 was 0.96, compared to a significantly lower AUC for CA 125 and HE4 alone. In fact, MUC16, when combined with several other biomarker candidates, such as CA19-9, EGFR, G-CSF, Eotaxin, IL-2R, MIF, and cVCAM, was reported to enhance combined sensitivity by more than 98% [193]. More recently, HE4 and MUC16 were analyzed using the ROMA model in females with an age cut-off of 51 years [194]. This study showed that ROMA performed best in females over 51 years old, whereas for females aged <51 years, a model of the combination of MUC16 and HE4 (with one or the other marker being positive) was superior with 100% sensitivity and more than 82% specificity. These studies establish MUC16 as a reliable marker for OC screening. Other mucins, such as MUC1 (CA 15-3), MUC2, MUC4, MUC5B, MUC13, and MUC17, have also been found to be upregulated during OC progression. Several efforts have been made to evaluate the diagnostic potential of these mucins in combination with other biomarkers, including a combination of MUC1 and MUC16 with other serum biomarkers, such as CA15-3, CA 27.29, and HE4 [116,186,195]. However, in most cases, mucin expression has been described based on the stage and molecular subtype of OC, which suggests that the mucin expression profile can be associated with stage-specific diagnosis and prognosis of OC [196]. Overall, mucins can be used to screen OC subtypes or design the combination biomarker panel. The results of serological analyses of mucin biomarkers in OC diagnosis and prognosis have been compiled in a table (Table 1-C). 

## 7. Colorectal Cancer Screening and Mucin Biomarkers

Colorectal cancer (CRC) is the second leading cause of cancer-related mortalities worldwide, accounting for 9.4% of cancer-related deaths [2]. However, an early screening and detection program that involves colonoscopy-based surveillance and removal of polyps has improved overall outcomes and significantly reduced CRC-associated mortality [197]. Colonoscopy is predominantly used for CRC screening [198,199]. However, recent advances in biomarker development for early-stage CRC detection have been appreciated as the liquid-biopsy-based biomarker assessment is non-invasive and can be performed frequently, unlike colonoscopy, where the adherence to follow-up is low [199,200,201]. For biomarker assessment, blood, stool, and urine samples are used to detect an array of markers, such as exosomes, circulating proteins, microRNAs, lncRNAs, and microbiomes [198,202,203,204,205,206,207,208]. 

Deregulated mucin expression has been observed during CRC progression [209,210], and mucins are thus considered as suitable biomarkers for risk assessment in early-stage CRC patients. Under normal conditions, MUC2, MUC4, and MUC17 are highly expressed in the colon, while other mucins, such as MUC5AC, and other glycoepitopes, such as CA19-9, are absent [210]. Various precursor lesions that are associated with CRC progression via distinct pathways (conventional or serrated), exhibit distinct mucin expression profile [210], which can be used to segregate patients to plan subtype-specific management of CRC. Furthermore, it has been observed that MUC2 and MUC4, which form the protective mucus layer in the normal colon, are gradually depleted during oncogenic progression. In colon carcinomas, these mucins have been reported to be entirely absent [41,42]. Therefore, analysis of MUC2 and MUC4 can be used to evaluate disease progression in CRC patients. Previous studies have shown that mucin expression in CRC depends on it’s origin. For instance, MUC1, MUC5AC, MUC17, and CA19-9 are expressed in carcinomas originating from the conventional pathway, whereas carcinomas originating from serrated pathways do not express these mucins [41,210]. Therefore, for early-stage detection, analysis of mucin expression profile could be a potential strategy for screening CRC patients or at-risk populations. Next, the expression of MUC5AC, CA19-9, and Tn/STn-MUC1 in polyps/hyperplastic polyps, which represent the precursor lesions of CRC, has been reported and can be used to stratify patients for routine surveillance and risk assessment [211]. Similarly, MUC2 and MUC5 can be used to evaluate the risk of lymph node metastasis in CRC patients [212]. 

Besides histopathological analysis, studies have shown that circulatory mucins can be used as biomarkers for CRC risk evaluation. For example, CA125 (MUC16), along with other serological markers, such as carcinoembryonic antigen (CEA), carbohydrate antigen 19-9 (CA19-9), tissue-polypeptide-specific antigen (TPS), and tumor-associated glycoprotein-72 (TAG-72), have been used clinically as serum-based biomarkers for CRC patients [119,209,213]. Similarly, MUC16, in combination with other glycoproteins, such as CEA, CA19.9, CYFRA21-1, and CA72-4, has been studied as a biomarker for the diagnosis and treatment guidance of CRC patients [119]. Previously, a serological study with an oncological 92-multiplex assay was used to correlate the serum expression profile of 92 immunological and oncological markers for disease prognosis. A total of 8 out of 92 markers in the panel, including amphiregulin, MUC16, kallikrein, IL-6, syndecan-1, TGF-α, and vimentin, were found to be significantly upregulated, and this panel could be used for CRC prognosis. On further analysis, high levels of MUC16 and serine protease kallikrein emerged as independent prognostic markers for CRC. Besides its tumor-promoting role in CRC, elevated MUC16 has been reported to promote metastasis and, therefore, is considered a marker for the evaluation of metastasis in CRC patients [214]. Circulating MUC1 has also been reported as an independent predictor of colon cancer. When combined with CEA, CA19-9, and cytokeratin-1 (CK-1), MUC1 emerged as a better biomarker for evaluating the risk of colon cancer than either of the markers alone [15]. Other mucins, such as MUC2, MUC4, and MUC5AC, can also be considered for designing a mucin-based panel for serum profiling or histopathological analysis of tissue biopsies for the screening of colon cancer (Figure 2). Previous studies that evaluated the diagnostic potential of mucins in early detection and disease prognosis are presented in the table (Table 1-D). As several other non-mucin proteins also deregulate during CRC progression, a combination of mucin and non-mucin proteins could be standardized as a biomarker panel to diagnose and stratify CRC patients in the early stage accurately. An unexplored area is the exploitation of knowledge of deregulated mucin expression for developing better optical probes for improving the efficiency and accuracy of screening. 

## 8. Mucins in Early Detection of Pancreatic Ductal Adenocarcinoma

Pancreatic ductal adenocarcinoma (PDAC) is one of the most lethal cancers of the gastrointestinal tract, with a five-year survival rate of 10% in the USA [3]. The lack of biomarkers to detect early-stage PDAC is a major factor contributing to the poor clinical outcome of existing treatment modalities for PDAC patients as most cases are diagnosed at advanced stage [215,216]. Pancreatic cancer is believed to originate from well-defined precursor lesions including pancreatic intraepithelial neoplasm (PanIN), intraductal papillary mucinous neoplasm (IPMN), and mucinous cystic neoplasm (MCN) [217,218]. Unfortunately, the asymptomatic nature of early PDAC makes it challenging to detect at an early stage. Several multidisciplinary efforts have been undertaken to develop imaging-based approaches, such as endoplasmic ultrasound (EUS) imaging, computed tomography (CT), and magnetic imaging resonance imaging (MRI), to detect early PDAC [219,220,221]. While pancreatic cystic lesions (MCNs and IPMNs) can be detected using various imaging modalities, imaging approaches cannot detect PanIN lesions, and imageable precursor lesions have a variable risk of developing into malignant disease. Thus, more sensitive detection approaches are needed in combination with existing approaches to enhance the detection sensitivity for early-stage pancreatic cancer. Among several putative biomarkers, mucins have been widely investigated for their potential use in detecting and analyzing PDAC progression, mainly due to their altered expression [16]. For instance, the mucin expression profile of a normal pancreas starts changing as soon as the early genetic and molecular events trigger the development of precursor lesions that further progress to PDAC [36,39,222]. The genomic and proteomic alterations in mucins have been studied in pancreatic tissues (both resected and EUS-FNAs), serum, and other body fluids of PDAC patients. The normal pancreas expresses MUC1, MUC6, MUC5B, MUC11/12, and MUC17, while other mucins, such as MUC2, MUC4, MUC5AC, MUC7, and MUC16, have not been detected [39]. Similarly, the mucins MUC20 and MUC21 are expressed at low levels in the normal pancreas, and MUC3 is expressed at the early stages of development, but its expression decreases after 13 weeks of gestation. An earlier report showed that MUC3 is heterogeneously expressed in the normal pancreas, but its expression is upregulated in PDAC along with other mucins, such as MUC1, MUC4, MUC5B, and MUC5AC [223]. Later, Park et al. reported a progressive increase in MUC3 PanINs with increasing dysplasia and high expression in PDAC [224]. Furthermore, MUC6 is expressed in the interlobular ducts, whereas MUC5B is expressed in the normal pancreas and the pancreatic duct. However, with the onset of a pathological insult, the expression, molecular modifications, and localization of mucins in the pancreatic tissue are altered. For instance, MUC1 expression changes from the central to the apical side of intralobular ducts, increasing as the disease progresses toward PDAC [16]. Furthermore, both PanINs and IPMNs are distinct in the mucin expression profile compared to the normal pancreas, and this differential expression profile could be used to stratify patients for designing treatment strategies [36]. In this regard, a previous study analyzed the expression of MUC1 and MUC2 mucins in PanINs and IPMNs [225]. This study showed that IPMNs (54%) predominantly expressed MUC2, whereas high-grade PanINs expressed MUC1 (61%) with infrequent MUC2 expression (<20%), suggesting that various precursor lesions have distinct mucin profiles. Another study showed that mucin MUC13 could differentiate IPMNs from non-mucinous cysts and therefore could be utilized in differentiating high-risk IPMNs from low-grade dysplasia [226]. Similarly, MUC5AC and MUC4 expression starts at the PanIN stage and increases progressively with disease progression [16]. These molecules could be used for early biopsy staining and serum profile analysis for detecting early stages of PC originating from these precursor lesions. 

Several studies have examined the utility of mucins, such as MUC1, MUC4, MUC5AC, and MUC16, for the detection of PDAC progression and prognosis [227,228,229]. Different biological samples, including tumor biopsies, serum, urine, and pancreatic secretions, have been analyzed to investigate the disease from PDAC patients alone or in combination with other biomarkers (Table 1-E). However, serum-based analysis of mucins is not used for the evaluation of diagnostic and prognostic approaches and the resectability of tumors in PDAC. As early PDAC progression drastically alters the molecular profile at the tissue and secretome levels, analyzing a multi-component biomarker panel might be more useful in early diagnosis than single-component analysis. For instance, CA19-9 is considered a gold-standard marker for PC disease management but has limited utility as a biomarker for early PC detection. However, when combined with mucins such as MUC16, CA19-9 has shown better sensitivity in differentiating early-stage PDAC patients from healthy controls [126,230]. Therefore, a combination biomarker panel is important for early PC diagnosis. Similarly, a combination of MUC16, CA19-9, and CEA was found to predict the benefits of postoperative adjuvant chemotherapy. In fact, low levels of these biomarkers correlated with improved surgical outcomes and prolonged survival of PDAC patients after adjuvant chemotherapy. Interestingly, another study highlighted high MUC16 levels in fibrotic tumors [231], suggesting that analysis of serum MUC16 could be used as a predictor of fibrosis in solid tumors, particularly the PDAC tumors that are highly fibrotic in nature. 

Secretory mucin MUC5AC has been extensively investigated as a biomarker in tissue biopsies and sera obtained from PDAC patients. Previously, two independent multi-center studies demonstrated that MUC5AC, in combination with CA19-9, performed better as a biomarker in PDAC patients than their individual performances [122]. Interestingly, MUC5AC was found to be expressed in early precursor lesions (PanIN1A/B) and the expression persisted with the disease progression, as analyzed in the samples from benign controls (BCs), resectable early-stage PC (EPC) patients, and unresectable late-stage PC (LPC) patients [122]. Analysis of serum samples showed that the circulating MUC5AC level in the EPC patients was significantly higher than that in healthy and benign controls. Notably, MUC5AC exhibited better sensitivity and specificity compared to CA19-9 as a single biomarker, and in combined analysis, MUC5AC enhanced the detection efficacy of CA19-9. Importantly, MUC5AC was found to increase the sensitivity (SN) and specificity (SP) of a combination biomarker (SN/SP; 67%/48% to 75%/83%, compared to CA19-9 alone) in differentiating EPC from BCs. Later, Zhang et al. analyzed MUC5AC and CA19-9 serum levels in PC patients (N = 61) and compared their combined efficacy in BC and CP patients and healthy controls. The combination of CA19-9 and MUC5AC showed a better performance as compared to the single biomarker, and the SN/SP of the combination biomarker was found to be superior to both biomarkers. However, these reports suggest that MUC5AC, in combination with CA19-9, is a reliable biomarker for the early detection of PC. Other mucins, such as MUC1 and MUC16, were also evaluated in combination with CA19-9 for PDAC diagnosis [232,233,234]. Interestingly, both MUC1 and MUC16 were found to complement the diagnostic efficacy of CA19-9. Similarly, the growth differentiation factor-15 (GDF-15), a TGF-β family member, has been reported to be a robust biomarker in differentiating between chronic pancreatitis and PDAC, and this efficacy was further improved when it was analyzed in combination with MUC16 [235]. We have compiled the information from previous studies investigating circulatory mucins for the early detection of PDAC (Table 1-E, Figure 1). These studies strongly support the notion that combination biomarker panels could be more clinically relevant for the early detection of cancer. Nevertheless, it would be pertinent to understand and differentiate between the expression profile of mucins in the serum of individuals with a normal pancreas, precursor lesions, and early-stage (resectable) PDAC, to select both positive and negative mucins in the detection panel. The circulating levels of tumor-specific mucins are likely to be very low in patients harboring precursor lesions or early disease and would thus require sensitive, reproducible, and high throughput assays to detect early-stage PDAC (Figure 2).

In the past decade, efforts have been made to develop more sensitive approaches for the detection of mucins in PDAC patient samples. For example, surface-enhanced Raman spectroscopy (SERS), magnetic gold-nanorod-based immunoassays, electrochemiluminescence, and mucin-targeted PET scans have been optimized to detect mucins in the ultrasensitive range. Previously, Wang et al. used SERS to detect MUC4 in the PDAC patient sera, outperforming conventional assays such as RIA and ELISA [236]. Later, a multiplexed-SERS-based immunoassay was developed to analyze the expression of CA19-9, MMP7, and MUC4 antigens in PDAC patient sera [237]. This study concluded that the combination panel used in the multiplexed SERS could differentiate between normal, chronic pancreatitis, and PDAC samples. Recently, a colorimetric immunoassay was developed using magnetic gold nanorods to detect CA19-9 and MUC1 [234]. Interestingly, the assay improved the detection limit for both the antigens, suggesting the clinical implication of this immunoassay for early detection of PDAC. In another multiplexed electrochemiluminescence assay, MUC16, CA19-9, HE, and CEA were analyzed in the sera obtained from healthy individuals and benign and PDAC patients [233]. This study showed that MUC16 significantly improved the performance of CA19-9 in discriminating between late- and early-stage PDAC from IPMNs. Combining these emerging biomarkers with imaging methods appears to further improve the performance of mucin-based PDAC diagnosis. Thus, combining mucin biomarkers with emerging technologies is a useful approach to improving PDAC early diagnosis.

## 9. Mucin Biomarkers in Prostate Cancer Diagnosis

Prostate cancer (PCa) is the most common malignancy in males. With an estimated 34,500 deaths in 2022, PCa remains the second leading cause of cancer-related deaths in males in the USA [3]. Compared to other cancer types, the 5-year survival of PCa patients is much better, mainly due to its symptomatic nature and slow progression, and extensive PCa screening programs. The most common screening test for PCa is the measurement of prostate-specific antigen (PSA) in the blood and digital rectal exam (DRE) [238,239]. However, it is sometimes hard to use the results of the PSA test for PCa diagnosis as factors such as older age, prostate enlargement, prostatitis, urologic procedures, smoking, and extensive workouts might raise the PSA levels [240,241]. Men with higher PSA levels often do not have PCa; therefore, further diagnostic screenings such as prostate biopsy are needed to confirm the diagnosis [238,242]. Therefore, identifying molecules or biomarkers that, in combination with available early screening tools, help improve the early detection of PCa and ultimately patient survival, is highly desired.

The progression of PCa subtypes is androgen-receptor-dependent, and several cancer-specific molecules, including mucins, are found aberrantly overexpressed in PCa [243,244]. It was reported in 1950s that the normal prostate secretions did not contain mucus; however, it was identified later that the mucoid type of secretions was noticed in post-atrophic hyperplasia (a precancerous condition) and well-differentiated prostatic carcinomas [245,246,247]. The association between aging, PCa, and changes in the mucin expression is of the utmost interest as a similar mucinous/mucoid metaplasia has been reported in the prostatic glands of guinea pigs with age and/or following the stilbestrol treatment [247,248]. Therefore, the expression pattern of mucins in PCa holds strong potential as both a diagnostic and prognostic biomarker. MUC1 has been comprehensively investigated and is reported to be aberrantly overexpressed in PCa [243,249]. The treatment of PCa cell lines and their xenografts in nude mice with MUC1 inhibitor decreased tumor progression and recurrence [243]. Additionally, MUC1 amplification or overexpression was associated with PCa relapse and bone metastasis [249]. MUC1 also regulates the plasticity of PCa subtypes such as castration-resistant prostate cancer (CRPC) and neuroendocrine prostate cancer (NEPC) subtypes [249,250]. In addition, MUC1 has been reported to promote NEPC progression and stem cell population in a E2F1-dependent manner [251]. The subtype-specific (CRPC and NEPC) roles and overexpression of MUC1 suggest a possible role for this molecule in identifying PCa subtypes and designing therapies. Similarly, the immunohistochemical analysis of PCa tumor tissues suggests that MUC5AC is overexpressed in the recurrent adenocarcinomas [252]. A recent study showed the overexpression of MUC1, MUC19, MUC4, MUC5AC, and MUC5B in the mucinous metaplasia of tissues isolated from Pten conditional knockout mice and human PCa tumor tissues [253]. High MUC5AC and MUC5B expression is associated with a high recurrence rate [253]. Further analysis suggested that the MUC4 expression is epigenetically silenced in PCa [244]. 

Expression of sialylated MUC1 in PCa patients treated with endocrine therapy was found to be increased, and its expression was associated with progression-free survival [128]. Interestingly, high serum MUC1 also contributes to patients’ response to androgen ablation therapy, and these patients are at higher risk of recurrence and visceral metastasis [129]. A comparative analysis of MUC1 and PSA in the serum samples of benign and malignant PCa patients suggested that high MUC1 is associated with the benign stage [130]. Similarly, an interesting study suggested a possible application of a highly sensitive aptamer-based immuno-loop-mediated isothermal amplification method for the early detection of MUC1 [254]. Overall, the implications of studies showing the differential expression pattern of various mucins in PCa can be extended to develop serological assays for the detection of circulatory mucins (such as MUC1, MUC5AC, and MUC5B) in PCa patients (Table 1-F). In addition, these mucin-based detection assays can be combined with the PSA screening method to develop a potential approach to improving the early detection of PCa and minimizing the chances of overdiagnosis from screenings performed considering PSA alone (Figure 1 and Figure 2).

## 10. Mucin Biomarkers in Liver Cancer

Liver- and intrahepatic-bile-duct-associated cancers are the third leading cause of cancer-related deaths worldwide and the fifth in the USA, with estimated annual deaths of 830,000 and 20,000, respectively [2,3]. Liver cancer is highly prevalent and a leading cause of cancer-related deaths in several transitioning countries in Asia and Africa, such as Mongolia, Thailand, and Egypt, where it is the leading cause of cancer-related mortalities. The liver cancer incidence rate is >2-fold higher in men than women, with common risk factors including hepatitis B and C viral infection (~54%), chronic alcoholism (~30%), diabetes, smoking, and other metabolic disorders (~16%) [255,256]. Early diagnosis favors a better prognosis in liver cancer patients. Imaging-based diagnostic methods such as ultrasonography, computed tomography (CT), or magnetic resonance imaging (MRI) are most commonly and reliably used to detect hepatic abnormalities, including malignant growth in the liver [257,258]. Additionally, several blood-based markers, such as alpha-feto protein (AFP), Des-γ-carboxyprothrombin (DCP), osteopontin (OPN), glypican-3, thioredoxin reductase, circulating microRNAs, α-L-fucosidase (AFU), and CA19-9, which have been characterized for early detection of hepatocellular carcinoma (HCC), have been thoroughly reviewed previously [259,260,261,262,263]. 

In the case of liver cancer, previous studies suggest a critical role of mucins in the progression of HCC and other liver-associated cancers [264,265,266,267,268]. For instance, MUC1 has been reported to be expressed during liver cancer progression, and the genetic knockdown or antibody-mediated targeting of MUC1 has been shown to inhibit its tumor-promoting functions in HCC [264,265,268,269,270]. Functionally, MUC1 has been reported to cooperatively interact with the c-MET receptor and regulate multiple downstream oncogenic pathways, including JNK-mediated phosphorylation of Smad2-, Bax-, and Caspase-8-mediated apoptotic pathways, and upregulation of activation protein-1 (AP-1) [264,265,269,271]. Similarly, MUC13 and MUC16 are overexpressed in HCC and other liver-associated cancers, promoting tumor growth and metastasis [266,267]. Furthermore, the significance of mucins as a biomarker and in predicting prognosis of HCC has been investigated previously. MUC1 expression was studied in preneoplastic lesions, fine-needle aspirates, and tissue biopsies derived from liver cancer patients, including cholangiocarcinoma (CC) and HCC [268,272,273,274]. These studies suggest that mucins such as MUC1 and MUC2 are absent in normal and preneoplastic foci but expressed in tumor biopsies, providing a prognostic significance of these mucins in HCC and cholangiocarcinoma. 

Serum levels of MUC1 and MUC16 have been thoroughly investigated as markers for HCC diagnosis and prognosis (Table 1-G). A retrospective study analyzed the serum MUC16 levels in HCC patients with different etiological histories of amebic hepatic abscess, chronic hepatitis, and acute viral hepatitis [132]. In all the HCC patients, the MUC16 levels were high, and it was concluded that MUC16 was a highly sensitive marker but with lower specificity for HCC. However, in another study, MUC16 was reported to be more sensitive than AFP (92% vs. 58.8%) in HCC patients at a cut-off value of 200 ng/mL. This study suggested that MUC16 could complement AFP in the diagnosis of HCC [133]. In a recent analysis, high preoperative serum MUC16 levels correlated with larger tumors and poor OS and PFS in HCC patients [138]. In another retrospective analysis of sera from 3440 HCC patients who underwent curative hepatectomy, the serum level of MUC16 was significantly higher in these patients than normal healthy individuals, suggesting that serum MUC16 could be used as a diagnostic marker for early HCC (Table 1-G). Moreover, a high MUC16 level has also been reported to correlate with younger age, gender, and an elevated AFP level. This study also suggested MUC16 as an independent prognostic factor of OS and PFS in HCC patients [134]. The potential role of MUC16 as a marker of HCC is further supported by another recent study by Qin et al., where sera from patients with hepatitis-B-virus-related HCC were analyzed for preoperative AFP and MUC16 levels [135]. This study showed that low preoperative MUC16 levels correlated with prolonged DFS and OS, while a higher baseline of MUC16 was found to be associated with poor prognosis. These studies clearly highlight that serum MUC16 could serve as a potential marker for HCC and could be explored for early detection of HCC and other liver cancers. 

Other studies have investigated the diagnostic and prognostic significance of serum MUC1 levels in liver cancers. Previously, based on a histological analysis that showed positivity of MUC1 in HCC and CC patients, MUC1 was analyzed in the sera of these CC and HCC patients [136]. MUC1 was significantly higher in patients with CC and HCC compared to the healthy controls, suggesting that MUC1 could be a marker for HCC and CC. Recently, sialylated MUC1 has been evaluated for disease recurrence in the sera of HCC patients who underwent radiofrequency ablation of primary tumors [137]. This study suggested that the serum sialylated MUC1 level was significantly higher in patients with disease recurrence and, therefore, considered an independent predictor of recurrence in HCC patients. Overall, serum mucin analysis can be useful for the early detection of liver cancer. However, studies are warranted to further explore mucins as biomarkers in liver cancer patients, and more importantly, it would be useful to correlate these findings with established biomarkers and imaging-based HCC diagnosis.

## 11. Mucin Autoantibodies for Early Cancer Detection

Autoantibodies are generated against tumor-associated antigens starting from early neoplastic development. Notably, tumor-associated (TA) autoantibodies are more abundant than their respective antigens and could be detected in cancer patients well before the tumor associated antigens accumulate to detectable levels [10,275], implying that the analysis of TA autoantibodies could be a valuable method for improving current early cancer detection approaches. Several studies have investigated the utility of TA autoantibodies in the early detection of epithelial malignancies such as breast cancer, lung cancer, and ovarian cancer [276,277,278,279,280]. In addition, there are reports suggesting the abundance of TA autoantibodies in PDAC, CRC, and prostate cancer patients [281,282,283]. Interestingly, TA autoantibodies against mucin antigens have also been detected in various epithelial cancers [284,285]. Previously, MUC16 autoantibodies have been detected in ovarian cancer patients. While the elevated MUC1 and anti-MUC1 antibodies exhibited prognostic significance in platinum resistant OC, anti-MUC16 antibodies showed no association [286]. In another study, risk of OC was assessed in patients with mastitis based on mucin autoantibodies [287]. Particularly, females with prior puerperal mastitis, caused by staphylococcus infection, had higher anti-MUC1 and anti-MUC16 autoantibodies, which was found associated with lower OC risk in these patients. This study suggested that long-lasting anti-mucin antibodies in mastitis associated with lower risk of OC. The diagnostic significance of anti-mucin autoantibodies for early-stage BC has also been explored. A previous study showed that analysis of anti-MUC1 autoantibodies are more reliable in ductal carcinoma in situ (DCIS) as compared to primary invasive carcinoma (PBC) of the breast. Compared to 18% grade III PBC patients, a total of 30% DCIS patients were positive for anti-MUC1 autoantibodies [288]. Later, an array of 61-mer MUC1 glycopeptide was used to detect autoantibodies in a larger cohort (n = 395 BC patients; n = 108 benign patients; and n = 99 healthy individuals). This study suggested that anti-MUC1 autoantibodies are higher in the case of early-stage BC compared to benign and normal cases. Moreover, the abundance of MUC1 glycopeptide autoantibodies was associated with reduced incidences and delayed metastasis [289], suggesting their diagnostic and therapeutic potential in BC patients. Circulating anti-MUC1 autoantibodies have been found both in the free form and bound to immune complexes and are associated with a favorable prognosis in early-stage BC patients [290,291,292,293]. Altered mucin glycosylation is associated with BC progression. A previous study suggests that O-linked glycosylation in mucins is common in BC patients and proposed it as a target for diagnosis and therapy [35]. Poza et al. recently developed a highly sensitive detection method for MUC1 autoantibodies using mimics of the MUC1-associated carbohydrate epitope Sialyl Tn (STn) for the detection of anti-MUC1 autoantibodies in BC patients [291]. Another report showed the presence of anti-MUC1 autoantibodies in the saliva and serum of HER2-positive early-stage BC patients [293]. 

Tumor antigen (TA) autoantibodies have also been investigated for early detection of LC [277,278]. In particular, the presence of anti-mucin autoantibodies in the circulation, either free or bound to immune complexes, is considered to have a high diagnostic value [294]. Similarly, efforts have been made to identify antigens in CRC patients that could be used to screen for TA autoantibodies [283]. Considering mucin autoantibodies as potential biomarkers in CRC patients, Pedersen et al. developed an array comprising of glycopeptides and glycoproteins corresponding to a MUC1, MUC2, MUC4, MUC5AC, MUC6 and MUC7 for seromic profiling of CRC patients [295]. The array exhibited a reasonably high specificity (92%) with good sensitivity (79%) and identified autoantibodies directed against aberrantly glycosylated peptides of MUC1 and MUC4. In a subsequent study, the same investigators evaluated the anti-mucin autoantibody signature in the women from UK Collaborative Trial of Ovarian Cancer Screening (UKCTOCS) cohort who developed CRC. The analysis included a microarray of synthetic glycopeptides of MUC1 and MUC4 bearing various carbohydrate epitopes to characterize the autoantibody signatures [284]. While autoantibodies directed against glycoepitopes exhibited some promising sensitivity in differential diagnosis of CRC, their performance in pre-diagnostic samples was limited. Further, MUC4TR5 autoantibodies significantly correlated with the risk of death in CRC patients, while a combination of MUC1 and p53 autoantibody signature was found to improve detection rates in pre-diagnostic samples. Mucin autoantibodies have also been found to be relevant in the early detection of PCa. Previously, MUC1 and its autoantibodies detected in the serum of PCa patients have been utilized for early cancer detection [128,130]. Recently, Somovilla et al. developed a highly sensitive fluorinated-proline-based MUC1 antigen for the detection of anti-MUC1 antibodies from PCa patient serum samples and reported that the detection of anti-MUC1 antibodies in PCa serum samples can be adapted for improved and early diagnosis of PCa compared to healthy individuals [131]. High circulating levels of anti-MUC1 IgG autoantibodies were associated with improved survival, while no association was observed with anti-MUC1 IgM. Despite their promising prognostic significance, in a comprehensive validation study, autoantibodies recognizingMUC1 glycopeptides or variable-number tandem repeat (VNTR) were found to be of limited utility for the early detection of PDAC, BC, LC and OC [296]. Overall, studies related to autoantibodies on one hand demonstrate the immunogenic nature of carcinoma-associated mucins and utility of autoantibodies as biomarkers, on the other hand these studies have also highlighted the challenges of using autoantibodies as biomarkers for early detection. However, designing panels to explore bioinformatically predicted immunogenic and neoantigenic epitopes in mucins and detailed characterization of the subclasses and isotypes against specific epitopes may help identifying promising autoantibody signatures.

## 12. Circulating Exosomes as a Cancer Biomarker 

The emerging role of exosomes in cancer pathogenesis has led to a paradigm shift in our conceptual understanding of extracellular vesicles and exosomes, which were previously considered carriers of cellular waste [297,298]. However, it is well established now that exosomes contain active cargos and are a primary resource for cellular communication, which is essential for proliferation, metastasis, and other pathological attributes of cancer cells [299,300]. As the exosomes are loaded with genomic and proteomic contents of the cell-of-origin, they serve as powerful surrogates for molecular profiles of cancers and are increasingly being explored for biomarker development. In the last decade, exosomes have been explored as biomarkers in different cancers, including lung cancer, colon cancer, breast cancer, and pancreatic cancer [298,299,301,302]. Interestingly, mucins have been found in exosomal fractions derived from liquid biopsies, suggesting that these mucins can be used identify and characterize tumor-specific exosomes. In a recent analysis of extracellular vesicles (EVs) derived from PDAC patients, different mucins, including MUC1, MUC4, MUC5AC, and MUC16, were detected in pancreatic juices [302]. In another study, analysis of EVs using a digital extracellular vesicle screening technique (DEST) reported the presence of a panel of mucins, including MUC1, MUC2, MUC4, MUC5AC, MUC6, and MUC13, in a cohort of 133 patients harboring intraductal papillary mucinous neoplasms (IPMN) [303]. Further, in the validation set, it was confirmed that MUC5AC is predominantly present in high-risk IPMN patients, suggesting that exosome-derived mucins could be potential biomarkers for early cancer detection. However, there are still challenges associated with the purification and characterization of exosomes from biological samples. Overall, mucin-packaged EVs and exosomes are potential resources for biomarker analysis in early-stage cancer detection. However, efforts need to be directed toward overcoming the technical challenges associated with exosome isolation and mucin biomarker analysis in these exosomes.

## 13. Conclusions and Future Perspective

Early detection and state-of-the-art therapeutic approaches are the key to better survival in cancer patients. However, current screening methods and therapeutic strategies are insufficient to constrain the increasing incidence of cancer-related deaths [3]. Oncogenic transformations are genetically guided, and so far, the understanding of factors associated with early mutations in different oncogenes is still limited. However, the subsequent molecular targets downstream of oncogenic mutations have been used to develop biomarkers and targeted therapies [304,305,306,307]. Due to their asymptomatic nature, most cancers are diagnosed at later stages when these are locally advanced or metastasized, and those patients are not considered eligible for surgical intervention. However, screening populations at a high or average risk of cancer has significantly improved the treatment outcome, quality of life, and survival in certain cancers, including ovarian, breast, and prostate cancer [3,308,309]. Despite this, current screening modalities have several limitations that need to be overcome. Therefore, early detection of cancers based on various biomarkers used in conjunction with screening techniques can play a pivotal role in the clinical outcome of cancer patients. Moreover, standard-of-care (SOC) systemic therapies and investigational drugs in clinical trials targeting the advanced disease stage tend to fail due to its aggressive phenotype, complex tumor microenvironment (TME), and acquired therapy resistance. Therefore, it is assumed that early diagnosis could provide a window and optimal targeting stage for these SOC investigational therapies, such as kinase inhibitors, anti-stromal therapies, and anti-angiogenic therapies [310,311]. Thus, biomarkers for early cancer detection are the need of the hour to improve cancer diagnosis, treatment outcomes, and patient survival.

Biomarker analysis is preferred for early cancer diagnosis, mainly due to its non-invasive nature and cost-effectiveness. However, the specificity and sensitivity of biomarkers are still challenging to achieve, and biomarker assessment often requires validation with cutting-edge imaging approaches. Therefore, novel biomarkers with high specificity and sensitivity are needed to enhance clinical acceptance. In this regard, mucin-based biomarkers have emerged as a potential target for early cancer detection. For example, CA125/MUC16 is an established biomarker for the screening and prognosis of ovarian cancer patients [179,189,312]. Previous studies on the assessment and validation of mucin expression in cancer patients’ sera and tumor tissues strongly suggest that mucin-based biomarker panels can be designed for early-stage cancer detection [41,57,62,313,314]. In addition, combining these mucin-based biomarkers with other bona fide detection approaches could provide more feasible, accurate, and sensitive diagnostic tools for early cancer diagnosis [122,230]. 

The structural, functional, and glycoproteomic characteristics of cancer-associated mucins have been utilized to differentiate between normal and early-stage cancer patients. Theoretically, genetic mutations, DNA methylation, splice variants, post-translational modifications, and their secretory nature can be assessed to compare and differentiate early-stage cancer patients from healthy controls [315,316,317]. Thus far, most studies have only investigated the expression of mucins at the tissue or serum level, the change in their glycosylation pattern, and their methylation status, which has often been found to be altered during the early stages of cancer progression [31,318,319]. Recent investigations have been more focused on identifying the disease-specific glycosylation pattern that shows a correlation with cancer progression [320]. Therefore, critical assessment of the glycosylation pattern of mucins could serve as an essential parameter for early cancer detection. In addition, combining serological mucin profiles with immune biomarkers could be another potential strategy for early cancer detection. Immunological parameters, such as immune infiltrates, the cytokine profile, and immune phenotypes, are considered critical in analyzing biological specimens for early cancer diagnosis [321,322,323]. In view of a previous study showing that increased glycosylation altered the binding of anti-MUC1 antibodies [34], we believe that it is important to analyze mucin glycosylation before considering a patient sample for biomarker analysis. 

Considering the success of CA125/MUC16 and other glycoproteins in early cancer detection and prognosis, it will be of interest to assess other mucin panels in larger cohorts of susceptible and at-risk populations to validate mucin-based biomarkers for early diagnosis of cancer. Thus far, most studies have used patient sera or tissue microarrays and performed retrospective analyses to assess and validate mucin biomarkers [41,230,324]. However, clinical practice is very limited in this area. Therefore, combining new-generation imaging technologies with mucin-based biomarker panels to accurately diagnose tumor development is still a challenge. In this regard, developing mucin-specific antibodies that are non-reactive to normal tissues is an ongoing challenge. Thus, the identification of cancer-specific mucin epitopes that could be used for generating selective and tumor-specific antibodies is required. Furthermore, several efforts have been directed toward correlating mucin-specific autoantibodies with cancer progression [289,291,305,325], which implies that the immunoassays used to detect mucin-specific autoantibodies might be useful platforms for early cancer diagnosis. However, the presence of autoantibodies must be correlated clinically to exclude inflammation and other pathogenic infections, which may also cause elevated levels of autoantibodies in patients. Another futuristic approach to developing biomarkers is serum- and tumor-derived ‘exosome’, a vesicular package of cancer-specific molecular entities that can be probed in serum and other biological fluids [326]. So far, only a few reports have highlighted the presence of mucins such as MUC1 and MUC5AC in cancer-associated exosomes [95,327]. Given that most cancers and precursor lesions exhibit a distinct mucin expression profile characterization of mucin in circulating exosomes can help identify early disease, risk prediction and stratification for mucin-targeted therapies. Overall, mucins have enormous potential as biomarkers, and combining new-generation approaches could provide a clinically relevant mucin-based biomarker(s) for early cancer detection.

## Figures and Tables

**Figure 1 cancers-15-01640-f001:**
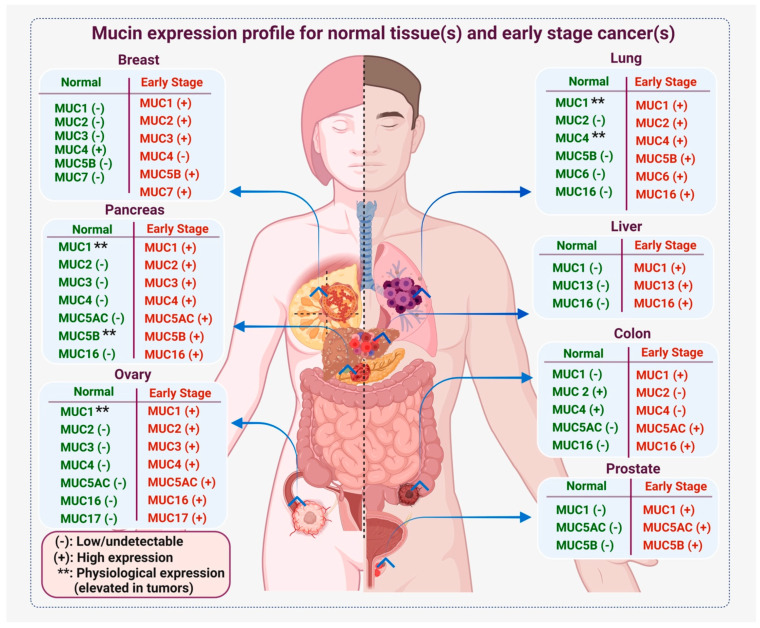
**Mucin expression profile in human malignancies.** Mucins are highly deregulated proteins in major human cancers. The **right panel** in the figure shows changes in the mucin profile in lung cancer, liver cancer, colon cancer, and prostate cancer, whereas the **left panel** shows mucin deregulation in breast, pancreas, and ovary cancers. A comparison of mucin profile of the normal and early malignancy is shown to emphasize their diagnostic potential in different malignancies. ** represents mucin expression under physiological conditions and the corresponding **(+)** sign represents further upregulation in cancer. The **(-)** sign shows non-detectable or lower expression under normal conditions and the **(+)** sign indicates the measurable increased/high expression of mucins. **Red:** early-stage expression profile; **Green:** mucin expression under normal conditions. The Illustrations in the figure were created with the help of BioRender tool.

**Figure 2 cancers-15-01640-f002:**
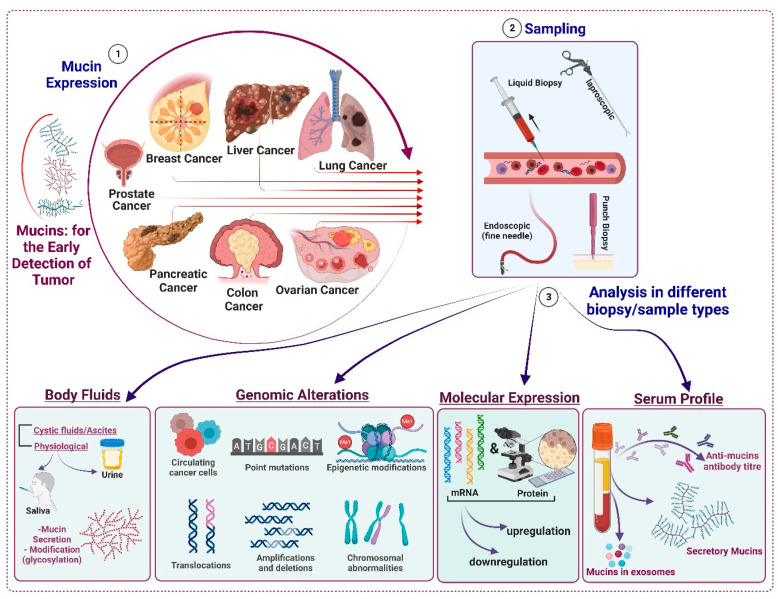
**Analysis of expression profile of mucins in cancer patients.** The genetic and epigenetic changes trigger oncogenic progression in different organs, which lead to the expression of disease-specific proteins including mucins. For early detection of cancers with mucin upregulation (as shown in **step 1**), liquid biopsies are collected using different methods (**step 2**). In **step 3**, the collected liquid biopsies are analyzed for genomic, transcriptomic, and proteomic profiling. Analyses of genomic alterations and molecular expression profiles are being increasingly used for detailed characterization of tumors. On the other hand, liquid biopsies such as serum and body fluids are considered convenient and suitable for the early diagnosis of cancers. These liquid biopsies can be evaluated for mucin expression and their post-translation modifications for early cancer detection. The Illustrations in the figure were created with the help of BioRender tool.

**Table 1 cancers-15-01640-t001:** Evidence for mucin-based biomarker analysis in various malignancies.

S. No.	Sample(s)	Mucin(s)/Combination	Stage of Detection	Readout	Ref.
**1-A: Lung cancer**					
1	Serum samples/ tumor tissues (IHC/ELISA), n = 80 samples	MUC16, IL6	NSCLC	High MUC16 and IL6 can be used to detect LC from liquid biopsies as they are positively associated with distant organ metastasis.	[101]
2	Exosomes from cell lines and patient plasma samples (n = 27 samples)	MUC1	NSCLC	MUC1 is enriched in the exosomes of cancer cell lines and patients (plasma) and can be used to detect NSCLC at an early stage.	[95]
3	Serum samples (n = 633)	CEA + CA125(MUC16) or CY211/NSE/SCC	NSCLC	The combination marker for NSCLC screening is CEA + CA125 (with positive cut-off range of 0.577CEA + 0.035CA125 ng/mL).	[106]
4	Serum samples (n = 289 suspected/unconfirmed, and n = 417 NSCLC, n = 96 SCLC)	CA125, CA19.9, CA15.3, TAG72-3, CYFRA21-1, CEA, SCC, NSE	NSCLC and SCLC	High CA125, CA15-3, CEA, CYFRA21-1, SCC, and TAG72-3 are early markers for NSCLC. High serum expression of CEA, TAG72-3, CA15-3, and CA125 denotes adenocarcinoma. High SCC, CEA, and CYFRA21-1 indicate squamous carcinoma.	[105]
**1-B: Breast cancer**					
1	Serum sample (n = 248 samples)	CA15-3(MUC1) and chemerin	Breast cancer and benign breast tumor patients	High chemerin with CA15-3 in the serum samples provided better diagnostic performance and could be used to characterize histologic grades.	[109]
2	Plasma samples (n = 200 BC patient, n = 47 benign breast lesions samples)	CEA and CA15-3	Breast cancer and benign breast lesions	High CEA and CA15-3 are seen in early-stage and cancer patients with node or distant organ metastasis.	[110]
3	Serum and tumor tissues (n = 433 samples)	MUC1	Primary and metastatic breast cancer tissues	High MUC1 in circulation/serum at the early stage can be detected in advance (two years before compared to other detection methods). Inclusion with multi-model screening strategies may diagnose tumors in women missed by mammography and irrespective of breast tissue density.	[111]
**1-C: Ovarian cancer**					
1	Serum samples (n = 46 stage 1 OC, n = 237 benign pelvic masses, n = 204 healthy controls)	CA125, M-CSF, and OVX1	Stage1 ovarian cancer, benign pelvic masses, and healthy women	A panel of CA125, M-CSF, and OVX1 tumor markers can identify early-stage ovarian cancer with extremely high sensitivity and moderate specificity.	[112]
2	Serum sample (n = 71 early-stage and n = 45 late-stage OC patients, n = 131 healthy controls)	OPN, MIF, IL8 AAb, and CA125	Early- and late-stage ovarian cancer patients	Combining OPN, MIF, IL8, and CA125 enhances the sensitivity of detection of OC patients compared to healthy controls.	[113]
3	Serum specimens (n = 75 invasive epithelial ovarian cancer and n = 547 healthy controls)	HE4 + CA72-4 and CA125	Pre-clinical invasive epithelial ovarian cancer and healthy controls	Combining HE4 + CA72-4 complements CA125 as a biomarker panel for longitudinal screening by multiplex assay.	[114]
4	Serum samples (n = 118 patients with ovarian tumors)	CA 15–3 and CA 27.29	Malignant and benign disease.	The serum concentration of CA 15–3 and CA 27.29 increased in malignant than in patients with benign disease.	[115]
5	Serum sample; n = 123 patients, either benign (n = 83 patients) or malignant (n = 40 patients)	CA15-3, CA27.29, and Panko Mab	Benign patients and malignant disease of the ovaries	PankoMab (anti-MUC1 antibody) has strong diagnostic potential in discriminating sera from patients with benign ovarian diseases vs. normal sera.	[116]
**1-D: Colorectal cancer**					
1	Serum panel (CEA, CA19-9, CK1, and MUC1) (n = 150 colon cancer, n = 50 benign, and n = 35 healthy controls)	CEA, CA19-9, CK1, and MUC1	Early-stage, benign, and healthy controls	Serum levels of CEA, CA19-9, CK1, and MUC1 gradually increase in benign disease to colon cancer compared to healthy controls.	[15]
2	Serum sample (n = 279 colorectal cancer patients vs. healthy controls)	CEA + CA19-9 + CA72-4 + CA125 + ferritin	Diagnostic potential and tumor status in CRC	Combining CEA + CA19-9 + CA72-4 + CA125 + ferritin has the diagnostic potential and evaluates the tumor status in colorectal cancer.	[117]
3	Serum and tissue sample (n = 22 CRC patients vs. healthy controls)	MUC1, MUC2	Early- and late-stage CRC patients	Increased MUC1 protein was observed in serum of late-stage CRC patients compared to control, whereas MUC2 was downregulated in CRC patients, as analyzed in tissue samples.	[118]
4	Patients (n = 373) with CRC evaluated pre- and post-surgery	CEA, CA19-9, CA125, CYFRA21-1, and CA72-4	Colorectal cancer patients	Combination of CEA, CA19-9, CA125, CYFRA21-1, and CA72-4 correlates with poor tumor differentiation and metastasis	[119]
5	Serum sample (n = 322 CRC patients vs. healthy controls)	CA125 and CEA	Colorectal cancer patients	CA125 turns out to be an independent prognostic factor in CRC with greater reliability than CEA.	[120]
6	Serum sample (n = 28 normal, n = 41 CRC/PC sera at 1 month, n = 33 CRC/PC sera at 2 months, and n = 25 CRC/PC sera at 3 months)	MUC5AC (NPC-1C Ab)	Colorectal/pancreatic cancer	MUC5AC(NPC-1C) antibody can discriminate the serum of cancer patients from normal donors in colorectal and pancreatic cancer.	[121]
**1-E: Pancreatic cancer**					
1	Serum and tissue samples (n = 346 samples)	MUC5AC, CA19-9	CP, early-stage resectable vs. late-stage non-resectable PC	Differentiates between early- and late-stage PC, and PC from CP.	[122]
2	Serum sample; early-stage PC (n = 30), late-stage PC (n = 31), 29 benign controls, 25 CPs, and 34 healthy controls	MUC5AC, CA19-9	CP vs. early PC And early vs. late PC, compared to healthy controls	Serum MUC5AC in patients with PC (210.1 (100.5–423.8) ng/mL) and combined biomarker panel (MUC5AC and CA19-9) showed a better performance.	[123]
3	Serum sample (n = 92 samples)	CA19.9, CA125, CEA, and CA242	Normal, benign, and PC	The combination panel enhanced the diagnostic efficiency.	[124]
4	Pancreatic juice (n = 191 samples)	MUC1, MUC2, and MUC4	IPMNs and PC	DNA methylation status differentiates between intestinal-type and gastric-type IPMNs.	[125]
5	Serum sample (n = 31 samples)	CA19.9, CA125, CEA, and CA242	PDAC patients undergoing cryoablation therapy	CA19.9, CEA, and TSGF for treatment assessment; CA242 for tumor staging, LN, and liver metastasis; TSGF for tumor differentiation.	[126]
6	Blood sample (n = 369 samples)	CA125 and CD4/CD8 ratio	Advanced-stage PDAC patients	Better prognosis with combined CA125 and CD4/CD8 ratio.	[127]
**1-F: Prostate cancer**					
1	Serum samples and tissues biopsy (n = 57 samples)	Sialylated MUC1 and PSA	Clinical stages and prognosis	Sialylated MUC1 increases with disease progression.	[128]
2	Serum samples (n = 11 patients)	Serum MUC16/CA-125	Tumor type or metastasis	PCa patient with elevated serum MUC16 (CA-125) had a high chance of persistent urinary symptoms and visceral metastasis.	[129]
3	Serum samples (n = 303 patients with benign and malignant disease)	Comparison of serum MUC1 and PSA	Benign vs. malignant disease	Elevated MUC1 (91% specificity) in benign disease, also high MUC1 in PSA-negative samples.	[130]
4	Serum samples	Serum anti-MUC1 antibodies and natural antigen for prostate	Early detection	High level of MUC1 antibodies in the early stages of PCa.	[131]
**1-G: Liver cancer**					
1	Serum samples of n = 115 HCC patients with a history of amebic hepatic abscess (62 patients), chronic hepatitis (40 patients), and acute viral hepatitis (41 patients)	Comparison of MUC16 and AFP serum level	HCC vs. benign hepatic disease	CA125 is a highly sensitive marker for HCC but lacks specificity.	[132]
2	Serum samples from HCC patients	MUC16 and AFP analysis	HCC vs. normal controls	MUC16 could complement AFP in diagnosing HCC. MUC16 is more sensitive than AFP (92% vs. 58.8%).	[133]
3	Serum samples of n = 3440 HCC patients underwent curative hepatectomy	MUC16 and AFP	Retrospective, HCC vs. normal individuals	Elevated preoperative MUC16 in 409 patients, correlated with younger age, females, and higher AFP level. CA125 served as an independent prognostic factor of OS and RFS.	[134]
4	Serum samples of n = 306 hepatitis-B-virus-related HCC patients	MUC16 and AFP	Preoperative HCC patients	A high MUC16 level was found to be risk factor for OS and DFS and correlated with the worst prognosis.	[135]
5	Serum sample from HCC patients (n = 27), CC patients (n = 8), metastatic liver cancer patients (30), healthy controls (n = 19)	MUC1 (KL-6)	Established HCC and CC	Significant differences in MUC16 levels of CC and HCC patients were seen compared to controls. All CC patients and 18.5% of HSS patients showed positivity above the cut-off (248 U/mL).	[136]
6	Serum sample of HCC patients (n = 144) who underwent complete radiofrequency ablation of primary HCC.	Wisteria Floribunda agglutinin (WFA)-positive sialylated MUC1	HCC patients after radiofrequency ablation	WFA-positive MUC1 correlated with HCC recurrence and was found to be associated with histological features of HCC.	[137]
7	Serum samples of n = 427 HCC patients with serum AFP level ≤200 ng/mL	MUC16 level analysis with cut off 30 U/mL	Preoperative serum analysis	CA125 levels were associated with maximum tumor diameter (>5 cm) and CA125 was found to be an independent risk factor of DFS and OS.	[138]

**Abbreviations: LC** = lung cancer; **NSCLC** = non-small-cell lung cancer; **BC** = breast cancer; **OC** = ovarian cancer; **PCa** = prostate cancer; **PC** = pancreatic cancer; **PDAC** = pancreatic ductal adenocarcinoma; **CA125** = cancer antigen 125; **CEA** = carcinoembryonic antigen; **CA19-9** = carbohydrate antigen 19-9; **MUC** = mucin; **IPMN** = intraductal papillary mucinous neoplasm; **MIF** = macrophage migration inhibitory factor; **OPN** = osteopontin; **M-CSF** = macrophage-colony-stimulating factor; **OVX1** = ovarian cancer antigen X1; **HE4** = human epididymis antigen 4; **CK1** = cytokeratin-1; **TSGF** = tumor-specific growth factor; **IL** = interleukin; **AAbs** = autoantibodies. **Analysis of mucins in liquid biopsies of cancer patients.** Mucins have been analyzed in liquid biopsies collected from different cancer patients. The table describes the stage of cancers in which mucins were detected, and major clinical readouts to demonstrate the potential of mucins as biomarkers.

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
