# Peer review of "Mucins as Potential Biomarkers for Early Detection of Cancer"

_cancers, 2023, doi:10.3390/cancers15061640_

Round 1

Reviewer 1 Report

The authors summarize the status of MUCIN family proteins in various cancers and discuss their potential as new biomarkers.

The paper is very well organized, and Fig. 1 and Fig. 2 provide a good overview.

The recent position of MUCIN as a biomarker is that it is not easy to simply differentiate between cancerous and normal counterparts by chemical staining, even in the case of carcinoma. It is important to be as non-invasive as possible (blood sampling is an acceptable limit), for example, as the authors note, the use of exosomes.

In the current situation, it is great to see the discussion of new perspectives.

Author Response

Reviewer 1: No specific comment.

We thank reviewer for reviewing our manuscript.

Reviewer 2 Report

This review article has been written professionally focusing on Mucins as potential biomarkers for the early detection of cancer.

In the abstract, the authors should highlight the keywords of this review article more. They should also write about the way they aim to report the literature. It is a little confusing to understand the content of the review only by reading the abstract.

There are only two paragraphs in the introduction. Do you think there is no more thing to add to the introduction because it is very short? How about adding a new paragraph to explain the available medicines for treating the different types of cancers? There are some patent reviews for the production of anti-cancer agents which are very expensive. I mean it is better to have early detection of cancer plus quick treatment with some anticancer agents.

https://link.springer.com/article/10.1007/s11030-022-10406-8

There are some subtitles that should be changed to a better ones. For example, by only reading "Lung Cancer" readers cannot understand the content of this part of the review article. The subtitles should be strong enough. 

Author Response

We thank reviewer for his comments. we addressed all the comments and following is the point wise response:

This review article has been written professionally focusing on Mucins as potential biomarkers for the early detection of cancer.

Comment: In the abstract, the authors should highlight the keywords of this review article more. They should also write about the way they aim to report the literature. It is a little confusing to understand the content of the review only by reading the abstract.

Response: Thanks for the comment. We have modified the abstract as per the reviewer’s suggestions.

Comment: There are only two paragraphs in the introduction. Do you think there is no more thing to add to the introduction because it is very short? How about adding a new paragraph to explain the available medicines for treating the different types of cancers? There are some patent reviews for the production of anti-cancer agents which are very expensive. I mean it is better to have early detection of cancer plus quick treatment with some anticancer agents.

https://link.springer.com/article/10.1007/s11030-022-10406-8

Response: Thanks for the suggestion; we have incorporated the reviewer’s suggestion in the updated manuscript.

Comment: There are some subtitles that should be changed to a better one. For example, by only reading "Lung Cancer" readers cannot understand the content of this part of the review article. The subtitles should be strong enough.

Response: Thanks for pointing out readers perspective regarding the sub-titles. We have modified each sub-title in the revised manuscript and included the “role of mucin biomarkers” in each cancer type.

Reviewer 3 Report

The review submitted by Gautam and colleagues gives an overview about mucins as biomarkers in cancer (lung, breast, ovarian, colon, prostate, liver and pancreas). The authors also provide a section about circulating exosomes.

The topic is interesting. There was previous review about mucins as biomarkers but any update is always welcome. However, the review is extremely dense and not always reader-friendly. I suggest improving the take home message by highlighting which mucin could be a good biomarker for each kind of cancer rather than a long list that does not help a lot.

11.       The authors state that autoantibodies and exosomes open new avenue of biomarker development. They should include a specific section about auto-antibodies (similarly as they did about exosomes)

22.       This review is enriched with the own publications from the Omaha group (more than 20). Some publications from other laboratories should also be cited such as Desseyn et al PMID: 18242885; Jonckheere et al PMID: 23178705;  Xu et al PMID: 35709153; Burchell et al PMID: 29903935…

33.       Lane 188.  The authors mentioned that MUC2 is not expressed during lung embryonic development. Why do they describe mucin expression during embryogenesis? This should be clarified.

44.       Lane 495. MUC11 and MUC12 designate the same gene. Please correct. (see https://www.genenames.org/data/gene-symbol-report/#!/hgnc_id/7510)

55.       There are several discrepancies between the figure 1 and the bibliography. For example, MUC1 is expressed in normal pancreas (mostly ductal cells). MUC5B was also shown to be detected. In the normal lung, MUC1 and MUC4 are expressed. MUC3 is not expressed in normal pancreas but there is no information about early stages.

66.       Does early stage designate precursor lesions (PanIN or IPMN for the pancreas) or cancer?

77.       The authors provide the mucin profile for lung cancer. Is there any difference between NSCLC and SCLC?

88.       Figure 2: mucin modification (glycosylation) is hardly a genomic alteration. It is post-translational

99.       In the table 1 (breast cancer), reference #138 is about indirect detection of MUC1 as they measured anti-MUC1 auto-antibodies.

110.   Table 1 (colorectal cancer), reference #299: only MUC1 is detected in the serum samples. MUC2 is evaluated by IHC in tumor samples.

111.   There are numerous mistakes in the references. I suggest to carefully going through the entire list. (I just saw several errors)

·         Reference #229 (about prostate) on lane 506 is cited for the MUC1/2/4 and IMPN.

·         Reference #166 (about ovarian cancer) is cited for MUCs in PanIN.

·         Reference #28 is the same as #281.

·         Reference #17 is the same as #57.

Author Response

We thank reviewer for critically reading the manuscript and for providing thoughtful comments. We have corrected the manuscript and made changes as suggested the the reviewer. Following is the point wise response to reviewers comments:

Reviewer 3:

The review submitted by Gautam and colleagues gives an overview about mucins as biomarkers in cancer (lung, breast, ovarian, colon, prostate, liver and pancreas). The authors also provide a section about circulating exosomes.

The topic is interesting. There was previous review about mucins as biomarkers, but any update is always welcome. However, the review is extremely dense and not always reader-friendly. I suggest improving the take home message by highlighting which mucin could be a good biomarker for each kind of cancer rather than a long list that does not help a lot.

Comment: The authors state that autoantibodies and exosomes open new avenue of biomarker development. They should include a specific section about auto-antibodies (similarly as they did about exosomes)

Response: As suggested by the reviewer, a special section for autoantibodies has been included in the revised manuscript (for details, please see the page#22 (line 723 to 783) of the revised manuscript).

Comment: This review is enriched with the own publications from the Omaha group (more than 20). Some publications from other laboratories should also be cited such as Desseyn et al PMID: 18242885; Jonckheere et al PMID: 23178705;  Xu et al PMID: 35709153; Burchell et al PMID: 29903935…

      Response: We thank reviewer for pointing out the issue and for suggesting some publications that were left out from the manuscript. Our group has been working for more than two decades on mucins. Therefore, there are good number of publications have been cited from our group. However, we included the relevant publications only.  Following reviewers’ suggestion and to make citations unbiased, we have included the suggested publications in the revised manuscript and thoroughly updated the citations.

Comment: Lane 188.  The authors mentioned that MUC2 is not expressed during lung embryonic development. Why do they describe mucin expression during embryogenesis? This should be clarified.

Response: Thanks for highlighting excess background related to MUC2 expression. We have removed it from the text in the revised manuscript.

Comment: Lane 495. MUC11 and MUC12 designate the same gene. Please correct. (see https://www.genenames.org/data/gene-symbol-report/#!/hgnc_id/7510)

Response: We thank reviewer for pointing out the mistake. We have corrected it in the revised manuscript.

Comment: There are several discrepancies between the figure 1 and the bibliography. For example, MUC1 is expressed in normal pancreas (mostly ductal cells). MUC5B was also shown to be detected. In the normal lung, MUC1 and MUC4 are expressed. MUC3 is not expressed in normal pancreas but there is no information about early stages.

Response: We appreciate reviewer for the critical reading of our manuscript.  As per the reviewer’s suggestions and available literature we have revised the figures.

Comment: Does early stage designate precursor lesions (PanIN or IPMN for the pancreas) or cancer?

Pancreatic cancer can originate from various precursor lesions including PanINs, IPMNs or MCNs, while resectable pancreatic cancer (Stage 1 and 2A) is generally regarded as early stage (Due to late clinical presentation of the disease) while locally advanced unresectable and metastatic PDAC is considered late stage. We have revised the manuscript to indicate the same.

Comment: The authors provide the mucin profile for lung cancer. Is there any difference between NSCLC and SCLC?

Response: We agree with the reviewer’s viewpoint for mucin expression in NSCLC and SCLC as the two cancers differs considerably; however very few studies are available on mucin expression in SCLC, so it is difficult to establish a clear difference among two cancer subtypes. Due to this reason, we have summarized the available information under single section of lung cancer.

Comment: Figure 2: mucin modification (glycosylation) is hardly a genomic alteration. It is post-translational

Response: As per the reviewer’s suggestions we have revised the figure 2.

Comment: In the table 1 (breast cancer), reference #138 is about indirect detection of MUC1 as they measured anti-MUC1 auto-antibodies.

Response: We thank the reviewer for the comment. We have included the word “autoantibodies” in the table to correct the statement.

Comment: Table 1 (colorectal cancer), reference #299: only MUC1 is detected in the serum samples. MUC2 is evaluated by IHC in tumor samples.

Response: We thank the reviewer for pointing this out; we have corrected the table accordingly.

Comment: There are numerous mistakes in the references. I suggest to carefully going through the entire list. (I just saw several errors)

  • Reference #229 (about prostate) on lane 506 is cited for the MUC1/2/4 and IMPN.
  • Reference #166 (about ovarian cancer) is cited for MUCs in PanIN.
  • Reference #28 is the same as #281.
  • Reference #17 is the same as #57.

Response: We are thankful to the reviewer for highlighting the errors in references. We looked references carefully and fixed the issues. We assure reviewers that references in the revised manuscript have been cited with no such errors. Please see the revised manuscript for updated bibliography.

Round 2

Reviewer 3 Report

It is very difficult to look at the revised manuscript without a track-change version.

It seems that the authors answered several of my comments

However, the figure 1 has not been corrected.

previous comment: There are several discrepancies between the figure 1 and the bibliography. For example, MUC1 is expressed in normal pancreas (mostly ductal cells). MUC5B was also shown to be detected. In the normal lung, MUC1 and MUC4 are expressed. MUC3 is not expressed in normal pancreas but there is no information about early stages.

In the figure 1: MUC1 and MUC5B still "not expressed" in normal pancreas (they are expressed!), similar comment for lung.

Author Response

Reviewer's comment: In the figure 1: MUC1 and MUC5B still "not expressed" in normal pancreas (they are expressed!), similar comment for lung.

Response: We sincerely thank the reviewer for the comment on the revised manuscript. We agree with the reviewer that MUC1 and MUC5B are expressed in the normal pancreas and lungs, further upregulating under cancer conditions. However, considering the biomarker point of view, we emphasized in the previous manuscript those mucins which can be used for cancer detection. Therefore, the (-) symbol was given to those mucins that are either not detectable or express low under physiological conditions, and the (+) symbol was for upregulated mucins.

However, considering the reviewer’s comment and to make the figure clearer and more understandable in the revised manuscript, we have further modified figure 1. In the revised figure, we have made changes suggested by the reviewer and included an (**) asterisk sign to mark mucin expression under the physiological condition in the pancreas, lungs, and other organs. In addition, we have modified text to describe mucin MUC3 and MUC5B expressions in pancreatic cancer. We hope this will address the discrepancies related to expression physiological expression of mucin.
